# Tunable Tensile Properties of Polypropylene and Polyethylene Terephthalate Fibrillar Blends through Micro-/Nanolayered Extrusion Technology

**DOI:** 10.3390/polym12112585

**Published:** 2020-11-04

**Authors:** Mahmoud Embabi, Mu Sung Kweon, Zuolong Chen, Patrick C. Lee

**Affiliations:** Multifunctional Composites Manufacturing Laboratory (MCML), Department of Mechanical and Industrial Engineering, University of Toronto, 5 King’s College Road, Toronto, ON M5S 3G8, Canada; embabi@mie.utoronto.ca (M.E.); mkweon@mie.utoronto.ca (M.S.K.); zchen@mie.utoronto.ca (Z.C.)

**Keywords:** micro-/nanolayered extrusion, nanofibrillation, nanofiber alignment, tensile properties, deformation ratio, shear stress

## Abstract

Fiber-reinforcement is a well-established technique to enhance the tensile properties of polymer composites, which is achieved via changing the reinforcing material concentration and orientation. However, the conventional method can be costly and may lead to poor compatibility issues. To overcome these challenges, we demonstrate the use of micro-/nanolayer (MNL) extrusion technology to tune the mechanical properties of polypropylene (PP)/polyethylene terephthalate (PET) fibrillar blends. PET nanofibers-in-PP microfiber composites, with 3, 7, and 15 wt.% PET, are first prepared using a spunbond system to induce high aspect-ratio PET nanofibers. The PP/PET fibers are then reprocessed in an MNL extrusion system and subjected to shear and extensional flow fields in the channels of the uniquely designed layer multipliers. Increasing the mass flow rate and number of multipliers is shown to orient the PET nanofibers along the machine direction (MD), as confirmed via scanning electron microscopy. Tensile tests reveal that up to a 45% and 46% enhancement in elastic modulus and yield strength are achieved owing to the highly aligned PET nanofibers along the MD under strongest processing conditions. Overall, the range of tensile properties obtained using MNL extrusion implies that the properties of fiber-reinforced composites can be further tuned by employing this processing technique.

## 1. Introduction

Micro-/nanolayer (MNL) coextrusion, a novel polymer melt processing technology that utilizes layer multiplication as a key step in the fabrication process, has recently attracted great interest in the polymer industry. While its operation is similar to that of conventional extrusion, multilayer coextrusion technology offers the advantage of fabricating polymeric structures with alternating layers, simply by installing a series of layer multipliers as an extension for the extrudate to flow through. As such, splitting and spreading of the polymer melt can produce multilayered products with individual layer thicknesses ranging from tens of nanometers to tens of millimeters. Recent studies exploited MNL coextrusion to manufacture films with unique optical, mechanical, and gas barrier properties. These films consist of hundreds of layers each less than 100 nm in thickness and include light-reflective films [1], optical lenses [2,3], and soundproofing film/foam multilayered composites [4,5].

Tangirala et al. fabricated novel light-reflecting multilayer polymer films with 1024 nanolayers in a strictly alternating fashion. Two different polymers with different refractive indices were employed and the individual layer thickness was controlled between 50 and 200 nm; consequently, the produced films exhibited pronounced optical interference effects [1]. Furthermore, Jin et al. constructed a synthetic lens using MNL coextrusion. By applying the layered concept encountered in biological lenses to nanolayer coextrusion, lenses consisting of polymeric nanolayered films with incremental differences in the refractive index were produced and molded into desired shapes. A lens with any refractive index distribution within the refractive index range of available extrudable optical materials was capable of being achieved [2]. In addition, introducing selective foaming to multilayered polymeric films provides further property enhancements such as sound barrier properties. Han et al. produced poly (vinyl chloride) based composites with alternating film/foam layers through a layer-multiplying coextrusion system. The experimental results revealed that the film/foam multilayered composites exhibit higher sound transmission loss and lower density than those of film/film composites [4].

Among numerous efforts to improve various films properties, enhancing the gas barrier properties of the coextruded composites through confined crystallization has been of particular interest in the field of polymer processing [6,7,8,9,10,11,12]. With an extended layer multiplication process, the resulting individual layer thickness diminishes exponentially and can reach down to tens of nanometers. This imposes a one-dimensional confinement on the crystal growth in the thickness direction, which forces spherulites to expand in a two-dimensional planar manner, thus increasing the tortuosity for gas diffusion. Additionally, MNL coextrusion can improve the tensile strength of multilayered composites compared to that of monolayer or bi-layer composites. Levitt et al., demonstrated that when an alternating stack of molten polymer sheets is uniaxially stretched in the direction parallel to the layers, the interfacial area per unit volume increases, thereby amplifying the interfacial effects [13]. The layer–layer interfacial reaction, present in nanolayered structures, was shown to play a vital role in enhancing extensional rheological and dielectric properties, due to the interface effect [14].

MNL coextrusion has also proven to be a rising technology in fabricating micro-/nanofibers. Recently, researchers have successfully obtained micro-/nanofibers by utilizing the MNL coextrusion technique, which is considered a nonconventional fibrillation method. Earlier studies have focused on combined concepts of horizontal and vertical layer multiplication to produce an extruded polymer–polymer composite with tunable domain sizes, which are controlled by the degree of process multiplication, exit die configuration, take-off method and post drawing process. After a suitable separation process, such as water jetting or solvation, nanoscale ribbonlike fibers are obtained from these extruded composites [15,16,17,18]. In addition, efforts have been made on in situ fibrillation using layer multipliers. In this processing technique, only one extruder and one-dimensional layer multiplication takes place, but the strong extensional and shear flow fields induced by the narrow channels within the multipliers orient and stretch the dispersed phase to transform its spherical/droplet morphology to fibrillar morphology. By varying the number of multipliers, different degrees of stretching are obtained and microscale fibers can be fabricated [19,20,21,22,23,24,25,26,27,28]. Moreover, He et al. investigated how the orientation of short glass fibers (SGFs) embedded in high-density polyethylene (HDPE) affects the tensile and impact strengths of composites created using varying numbers of layer multipliers. The study revealed that the dispersed SGF tend to orient more along the flow direction with added multipliers, which immensely improves fracture toughness and crack propagation resistance by bridging the crack propagation in the direction perpendicular to oriented SGFs [23].

Another processing technique to fabricate polymer composites with enhanced properties is in situ nanofibrillation, which is a technology that produces nanofiber-in-microfiber composites that are considered as new members to the polymer composite family. Fakirov et al., outlined a number of parameters that must be considered when manufacturing micro-/nanofibrillar composites [29]. First, the chosen materials should lead to an immiscible blend with the minor phase adopting discrete droplet/spherical domains before stretching. Next, the melting point of polymer B should be at least 30 °C higher than that of polymer A to allow a wide processing window in further processing operations and to avoid damaging the fiber morphology of polymer B. Lastly, a large extensional force to deform polymer B droplets into fibers must be present. Many studies exploring this technology and how it affects the properties of the composites have emerged recently [30,31,32,33,34,35,36,37,38,39]. Zhao et al., investigated different grades of polypropylene (PP) matrix in which polyethylene terephthalate (PET) nanofibers were dispersed. It was concluded that a matrix with lower melt flow rate (MFR) was more advantageous for generating PET nanofibers with a high aspect ratio and smaller diameter across a broad filler concentration range [38]. Rizvi et al., also fabricated a PP/PET nanofibrillar composite with not only enhanced mechanical properties, but also significantly enhanced foamability. These results revealed that the presence of PET fibers in PP matrix yields foams that exhibit up to two orders of magnitude higher cell densities and up to a five-fold increase in the expansion ratio relative to those of the neat PP [39].

Existing studies on micro-/nanofibrillation and MNL coextrusion have provided a strong impetus for integrating both manufacturing platforms to ultimately engineer multiphase nanostructured composites with enhanced properties. While several research efforts have focused on (1) in situ formation of microfibers via droplet morphology deformation within layer multipliers, and (2) fabrication of dual-component fiber mats using two-dimensional multiplication, characterizing the relationship between mass flow rate and the degree of orientation of high aspect ratio, prespun polymer nanofibers within the layer multipliers remains unexplored. Therefore, in this work, we take advantage of the narrow channel design of the multipliers and high elongational flow to obtain highly oriented PET nanofibers along the flow direction. This preparation procedure provides a solution to fabricate polymer matrices reinforced by nanofiber polymers with different degrees of fiber alignment to tune the tensile properties. Firstly, a labscale spunbond system was used to obtain PP/PET nanofiber-in-microfiber blends containing 3, 7, and 15 wt.% PET contents. The diameters of the PET fibers obtained from the spunbond process were as low as 50 nm, and with average values of 135, 139, and 192 nm for 3, 7, and 15 wt.% PET concentrations, respectively. The fibrillar blends were then re-fed into an MNL extrusion system with a different number of multipliers. The processing temperature throughout the extrusion stage was maintained above the melting point of the PP matrix and below that of the dispersed PET phase to prevent the PET nanofibers from melting. The change in orientation of PET nanofibers was observed under varying numbers of multipliers and mass flow rates. Neat PP was processed and tested under the same conditions as a basis for comparison. By strengthening the extensional and shear flow fields through increased mass flow rate and number of multipliers, we aim to elucidate how the orientation of the PET nanofiber network on a microscopic scale enhances the mechanical properties of the composite.

## 2. Materials and Methods

### 2.1. Materials

The matrix polymer employed in this study is a PP commercially available as ExxonMobil PP3155 (weight average molecular weight = 188,000 g mol^−1^; polydispersity index = 4.2) (ExxonMobil, Irving, TX, USA) with an MFR of 36 g/10 min (230 °C/2.16 kg) and a melting temperature of 165 °C. PET used as the reinforcement material was kindly donated by Lotte Chemical Corporation (Seoul, Korea), commercially available as HOT^TM^; its melting temperature is 253 °C, and the intrinsic viscosity is 0.8 dL/g.

### 2.2. Fibrillation of PP/PET Blends

PP and PET granules were dried for 8 h in a vacuum oven at 80 °C and 100 °C, respectively, dry blended at 97/3, 93/7, and 85/15 wt.% compositions, and fed into the hopper of a lab-scale spunbond system. As shown in Figure 1, the spunbond system comprises of multiple components, among which a twin-screw extruder, with a screw diameter of 20 mm and a length-to-diameter ratio of 32:1, and a circular flat die (referred to as the spinneret), are mounted on an elevated stage. Polymer strands exiting the spinneret are pulled down by an air suction force using an aerodynamic device known as the air drafter. This pulling force stretches the molten PP/PET blend and transforms the spherical PET domains into nanofibers to ultimately produce a nanofiber-in-microfiber morphology. The stretched fibers are then collected on a moving belt driven by a roller, creating a nonwoven fiber mat as an end product of this process. To ensure complete melting of both phases, the zone temperatures from the hopper to the spinneret were set as follows: 220 °C, 260 °C, 260 °C, 260 °C, 260 °C, and 275 °C. The feeding zone was 40 °C cooler than the barrel temperature due to a water cooling system installed in this region of the spunbond machine, while the die was set 15 °C higher to further reduce the melt viscosity in the final zone before air stretching. Moreover, to maximize the capability for fiber stretching, the throughput of the extruder should be minimized and the air suction should be maximized. Thus, the feeding frequency and screw rotating speed were set to a minimum at 16 g/min and 95 rpm, respectively, while the air drafter speed was set to a maximum at 20 m/s.

### 2.3. MNL Extrusion

The first step in layer multiplying coextrusion employed in this work involves two polymer streams combining in a feedblock to create a three-layered extrudate as they exit, in an A-B-A format. The layered stream then enters a series of layer multipliers, also referred to as *laminating multiplying elements (LME’s)*. Each multiplier splits the stream vertically in half, diverts both half-streams into two separate channels, and re-stacks them on top of one another as they exit, while maintaining a constant width and thickness (Appendix A). Owing to repeated dividing and stacking of layers, layer multiplication is an exponential process where *n* multipliers result in 1 + 2^*n*+1^ layers in total.

However, rather than creating alternating structures from two different polymer streams, the main objective in this study was to induce preferential orientation in the PET nanofibers by applying strong extensional and shear flow fields through the narrow channel design of the layer multipliers. For this reason, only one extruder (Leistritz, Nurembourg, Germany) was used in this processing setup. The nonwoven fiber mat roll was mounted on a horizontal shaft and gradually fed into a Leistritz corotational 27 mm twin screw extruder (length-to-diameter ratio of 40). A series of layer multipliers were assembled to the end of this extruder (Appendix A). Differential scanning calorimetry (DSC 250, TA Instruments, New Castle, DE, USA) was used to obtain the heat flow curves of the PP/PET fibrillar blends with all three PET concentrations (Appendix A). Based on the thermograms, the temperature zones along the extruder and the multipliers were set to 200 °C—above the melting point of PP but below that of PET—to preserve the PET fibrillar morphology within the molten PP matrix during processing. In order to investigate the effect of mass flow rate, the screw speeds were set to 50 rpm, 100 rpm, and 300 rpm to produce mass flow rates of 40 g/min, 85 g/min, and 245 g/min respectively.

In addition to the effect of mass flow rate, the influence of layer multiplication was explored. The layer multipliers were designed to provide convergence in the thickness direction followed by extension in the planar direction, to allow layer stacking at a constant overall thickness and width, respectively. PET nanofibers flowing in this polymer stream undergo deformation as a result of this channel design. This deformation ratio is dependent on the number of layer multipliers used in the process, and is estimated using the method adopted in Li et al.’s study [21]. As shown in Figure 2, the deformation ratio is a function of the number of multipliers and is equal to the product of both the converging ratio along the thickness direction (xz-plane) and the extending ratio along the planar direction (xy-plane). By assembling six multipliers in series, the deformation ratio can reach up to 2^12^, on the order of 10^3^.

Moreover, the shear stress experienced in the multipliers was estimated from a non-Newtonian flow simulation using COMSOL^®^ Software (COMSOL AB, Stockholm, Sweden). The simulation was conducted using the power law model of PP at 200 °C and was repeated for each mass flow rate. The rheological data of the PP matrix at 200 °C (Appendix A) was obtained using a rotational rheometer (ARES-G2) from TA Instruments (New Castle, DE, USA), and the fluid consistency coefficient and the power law index were estimated at 618.3 N*s^0.771^/m^2^ and 0.771, respectively.

For each PP/PET blend, six different combinations of multipliers and mass flow rates were explored (Table 1). Each combination was coded with the following format: *a*%-*b*M-*c*, where *a*, *b*, and *c* indicate percentage of PET concentration, number of multipliers, and screw rpm, respectively. Neat PP subjected to the same thermal history as that of the composites was processed under zero multipliers and 50 rpm (40 g/min) and used as a control sample.

### 2.4. Scanning Electron Microscopy (SEM)

Quanta FEG 250 SEM (Thermo Fisher Scientific, Waltham, MA, USA) was used to characterize the morphology of the composites and orientation of the dispersed phase. To analyze the size distribution of PET spherical domains in the PP matrix prior to fibrillation, the samples were placed in liquid nitrogen and cryogenically fractured. To analyze the PET nanofiber size distribution after the fibrillation process, the PP/PET-fiber composite samples were first etched in boiling xylene for 3 h to remove the PP matrix. The remaining PET fibers were then dried in an oven to remove the xylene. Finally, to analyze the PET nanofiber orientation in composites obtained from MNL extrusion, the samples were cryogenically fractured parallel to the flow direction. The fractured surface was partially etched in boiling xylene, for 1 h at 140 °C, to expose the PET fibers while maintaining their orientation in the matrix. All samples were sputter coated with platinum prior to SEM characterization. The spherical PET phases and nanofiber diameters were measured using ImageJ software (National Institutes of Health, Bethesda, MD, USA). For statistical accuracy, 200 measurements were performed for each condition.

### 2.5. Orientation Angle Measurements

To understand the processing-structure relationship, SEM was employed to observe and measure the angle of orientation along the MD from the samples processed via MNL extrusion using different number of multipliers and mass flow rates. The orientation of PET fibers with respect to the machine direction (MD) was quantified similar to the method illustrated in Figure 5 in He et al.’s study [23]. The angle between a nanofiber strand and the MD was measured using the Angle Tool in the ImageJ software (Appendix A) (National Institutes of Health, Bethesda, MD, USA). All measurements were recorded as positive angles between 0° and 90°. The distribution of nanofiber orientation was obtained from 100 angle measurements, from which the average orientation angle and the standard deviation were calculated. The average orientation angle served as a measure that represented the overall degree of nanofiber alignment, where 0° indicated perfect alignment of fibers parallel to the MD, 90° indicated orientation of fibers transverse to the MD, and 45° implied random orientation. In addition, the standard deviation was used to characterize the dispersion of nanofiber orientation; the smaller the value, the more fibers were preferentially aligned in a particular direction. Three samples were used for each PET concentration: 0M-100 “low extreme”, 6M-100 “intermediate”, and 6M-300 “high extreme”. The low extreme and intermediate cases have the same mass flow rate of 85 g/min (screw speed 100 rpm), while the number of multipliers increases from 0 to 6, to test the effect of added multipliers on the orientation angle. Moreover, the effect of increasing the mass flow rate is investigated by comparing between the intermediate and high extreme cases, as the number of multipliers is maintained at six, while the mass flow rate increases from 85 to 245 g/min (screw speeds 100 and 300 rpm, respectively).

### 2.6. Mechanical Test

The mechanical properties were characterized using an Instron model 5965 tester (Norwood, MA, USA). The tensile samples were prepared by compression molding, at 200 °C and 1000 psi for 10 min, using an ASTM D-638 Type 5 mold. Samples were placed on the mold in a way that the MD was parallel to the axis of the tensile bars. The test conditions were set according to ASTM D-638 with a cross head speed of 5 mm/min. At least five samples were tested for each condition and an average value was recorded.

## 3. Results

### 3.1. Size Distribution Analysis

The mechanical properties of any polymer composite depend significantly on dispersion, distribution, and the size of the dispersed phase. Morphological analysis was conducted on the PET dispersed domains with all concentrations in the prespunbond, postspunbond, and post-MNL processing stages. Figure 3A1–A3 shows the PET phase in droplet morphology. The micrographs indicate uniform dispersion and distribution of the PET in the PP matrix for each individual PET concentration. Figure 4A1–A3 shows the size distributions of the PET droplets in the PP matrix for all PET concentrations. As expected, a gradual increase in the average PET droplet size was observed with an increase in its content from 3 wt.% to 15 wt.%. The finest PET spherical domain obtained was 2.78 ± 0.77 µm at a 3 wt.% PET loading. On the other hand, the poorest PET dispersion was obtained using 15 wt.% PET, resulting in an average diameter of 6.21 ± 1.89 µm. This observation is consistent with the increased domain size from more droplet-droplet coalescence occurring as the dispersed phase concentration increases in immiscible polymer blends [40].

Figure 3B1–B3 shows the morphologies of the PET nanofibers after the fiber stretching process in the spunbond system, and the respective diameter distributions are shown in Figure 4B1–B3. Despite the droplet coalescence phenomenon that resulted in a droplet size increase seen in the prefibrillation step, the postspunbond PET nanofibers showed more uniformity in average diameters across the range of PET concentrations studied. Blends with 3 and 7 wt.% PET showed similar distribution graphs with an average diameter value of 0.14 µm and respective standard deviations of 0.06 µm and 0.03 µm. The average nanofiber diameter for 15 wt.% PET was 0.19 ± 0.05 µm, showing a relatively slight increase in size compared to the increase recorded in the droplet size distributions. The fine nanofiber diameters obtained across all PET concentrations indicate good fibrillation achieved during the spunbond process, in which the deformation mechanism stretching droplets was more dominant rather than droplet coalescence. Moreover, by looking at the SEM images in Figure 3C1–C3 and the respective diameter distributions in Figure 4C1–C3, one can observe little to no change in nanofiber size distributions post-MNL process compared to the postspunbond stage, indicating that the MNL processing stage did not deform the PET fibers and preserved the fibrillar structures as intended. Table 2 summarizes the average diameters and standard deviations of PET droplets and fibers under different PET concentrations and processing stages.

### 3.2. Orientation Angle Analysis

The SEM images of each sample tested for nanofiber orientation are shown in Figure 5. The images reveal that the PET fibers gradually become more oriented along a preferential direction (MD; horizontal as indicated by the arrow) from the low extreme to the intermediate, and finally to the high extreme cases. This trend is consistent in all three PET concentrations studied. However, the effect of increasing mass flow rate seems more effective as a clearer stretch and orientation of fibers was observed as the mass flow rate increased from 85 to 245 g/min (i.e., intermediate to high extreme). This is the result of more intense extensional and shear flow fields induced in the flow channel as more material is pumped through a small cross section, making it a more dominant factor in aligning the nanofibers along the MD.

Figure 6 shows the corresponding angle of orientation distributions for each of the tested samples. Samples of A1, B1, and C1, which belong to the low extreme case, revealed randomness in their respective distributions, with average orientation values of 41.0°, 44.9°, and 37.9°, respectively. The high standard deviations also indicate that a very weak nanofiber orientation is induced under such moderate processing conditions during MNL extrusion. Samples A2, B2, and C2, show the distributions after introducing six multipliers to the MNL process. All three orientation angle distributions show denser clusters shifting closer to zero. As such, the average orientation angle values significantly decreased indicating a higher degree of orientation. This observation is in accordance with the results obtained by He et al., [23] who demonstrated that increasing the number of multipliers lowers the average orientation angle and standard deviation of SGFs. Moreover, samples A3, B3, and C3 reveal the distributions after increasing the mass flow rate from 85 g/min to 245 g/min. The average orientation angle values decreased even further, and the significantly lower standard deviations indicate that the majority of PET fibers are aligning parallel to the MD. The distributions show more concentrated clusters closer to zero compared to samples A2, B2, and C2. The average orientation values and standard deviations are listed in Table 3. Overall, the qualitative and quantitative analyses provide clear evidence of increased nanofiber orientation along the MD as a result of increased number of multipliers and mass flow rate, demonstrating that MNL extrusion technology is capable of providing nanofiber-in-microfiber polymeric composites with further oriented nanofibers.

### 3.3. Tensile Testing

The tensile performance of fiber-reinforced composites highly depends on the fiber content aligned with the loading direction. By increasing the mass flow rate and number of multipliers, anisotropic orientation of the PET fibers along a preferential direction (i.e., MD) was created. Controlled tensile tests, in which the loading direction is parallel to the MD, were conducted with respect to the processing conditions. The measured average elastic modulus, tensile strength, and ductility for all PP/PET blend compositions are shown in Figure 7. The test samples were chosen to study the effect of two main factors: (i) number of multipliers (i.e., deformation ratio) and (ii) mass flow rate (i.e., shear stress). Neat PP and undrawn blend specimens produced under zero multipliers and 85 g/min flow rate were tested as a control.

Firstly, four samples processed under zero, two, four, and six multipliers at a constant mass flow rate of 85 g/min were tested. In Figure 7a, with zero multipliers assembled, the elastic modulus of the PP/PET blend with 3 wt.% PET was measured to be 1381 MPa. This recorded value shows little PET nanofiber reinforcement, which is attributed to the random PET nanofiber orientation. The elastic modulus showed a slight and steady increase, from 1381 MPa to 1448 MPa as the number of multipliers increased from 0 to 6, resulting in a final increase of 5%. This property enhancement is more pronounced with 7 wt.% PET, where a 10% increase was observed, showing the greatest margin of increase of all PET concentrations studied. On the other hand, little to no change was observed in samples with 15 wt.% PET as a result of increase in number of multipliers. Furthermore, Figure 7c shows recorded margins of increase, for tensile strength, of 6%, 13% and 14% for 3, 7, and 15 wt.% PET, respectively. Surprisingly, samples with 15 wt.% PET resulted in consistently lower values of tensile strength than those of other two PET concentrations across all processing conditions. This phenomenon may be attributed to the interfacial slip between the PP matrix and dispersed PET phase due to an excessively increased interfacial area between the two immiscible polymers at a higher PET content, which, in turn, may serve as a greater source of weakness in multiphase solids [41,42,43]. Similar behavior was observed in earlier fibrillation studies [30,38,44], and further research on improving the interfacial bonding of this immiscible polymer system is necessary to further enhance the mechanical properties of PP/PET composites. Weak interfacial area between the matrix and dispersed phases resulting in poor translation of applied stress onto the composite is also revealed in Figure 7e, which shows that the elongation at break decreases at higher PET concentrations, illustrating the effect of poor interfacial adhesion. In fact, the elongation at break did not show any trend with respect to the number of multipliers.

Secondly, the effect of mass flow rate was studied across the following samples; 6M-50, 6M-100, and 6M-300 for all PET concentrations. Figure 7b shows that the elastic modulus also increased with increasing mass flow rate. Increasing the mass flow rate from 40 to 245 g/min improved the modulus by 13.5%, 19.3%, and 35.0% for 3, 7, and 15 wt.% PET, respectively. Similarly, Figure 7d shows a tensile strength increase of 8%, 9%, and 20% for 3, 7, and 15 wt.% PET, respectively, as a result of the same increase in mass flow rate. Such a tremendous property enhancement can be ascribed to the improved orientation of nanofibers along the MD during the MNL extrusion process. Processing the composite under the maximum number of multipliers and highest mass flow rate shows the highest degree of orientation of all samples, as shown in Figure 5 and Figure 6. In addition, the strongest processing condition not only produced highly oriented PET nanofibers that serve as heterogeneous PP crystal nucleation sites but also promoted flow-induced crystallization of the PP matrix [38], resulting in synergistic effects on improving the mechanical properties. Hence, the effect of mass flow rate is crucial to enhancing tensile modulus and strength. Moreover, little to no change was observed in the ductility of the composites, as shown in Figure 7f.

Furthermore, given the effect of PET nanofiber orientation on the tensile behavior of each PP/PET blend, the change in tensile properties as a result of average angles of orientation provides a vivid understanding of the structure-property relationship. The variations of elastic modulus, tensile strength, and ductility with respect to average angles of orientation were plotted for all PP/PET blends studied, and illustrated in Figure 8. From random alignment along the MD to high alignment, the increase in Young’s modulus is about 16%, 27%, and 35% for 3, 7, and 15 wt.% PET, respectively, as shown in Figure 8a. A similar observation was also made for tensile strength, shown in Figure 8b. A sharp increase in tensile strength can be seen for 3, 7, and 15 wt.% PET with margins of 16%, 15%, and 30%, respectively. Such increase is significantly greater than that reported by Zhao et al. who used the same PP matrix to produce PP/PET-fibril composites with a 6% increase in tensile strength at their optimal PET concentration of 6 wt.% [38]. In addition, the tensile strength was reported to decrease as the PET concentration increased beyond this critical loading, as reported in other fibrillar composites [30,44]. Hence, it is clear that the MNL extrusion process leads to PET nanofibers that induce an amplified reinforcing effect when aligned along the loading direction [22], as in any fiber-reinforced composite. In specimens with average angles of orientation closer to 45°, a larger amount of the tensile deformation applied to the weak matrix is translated to more randomly dispersed PET fibers, resulting in low stiffness and tensile strength. On the other hand, as the number of aligned PET fibers along the loading direction increase, the tensile load is transmitted more to the reinforcing fibers creating a pronounced reinforcing effect. Moreover, elongation at break, as a measure of ductility, indicates slight variations with average angles of orientation, as illustrated in Figure 8c. As more PET fibers deviate off the MD indicating a more random orientation, the failure is more dominated by the matrix and interfacial properties. Therefore, deformation within the matrix is dominant as the average angle of orientation shifts closer to 45°, resulting in relatively higher ductility. Overall, integrating the processing platforms of nanofibrillation and MNL extrusion has shown to have a huge potential to tune the mechanical properties of fiber-reinforced polymer composites. With simple setting adjustments, the tensile properties of a specific PP/fibrillar-PET blend can be enhanced, overcoming the need to increase the concentration of a reinforcing material.

## 4. Conclusions

Advanced MNL extrusion and nanofibrillation technologies were successfully integrated in this study. The nanofibrillation spunbond system yielded uniform high aspect ratio PET nanofibers, as low as 0.055 µm in diameter, which were systematically reprocessed in the MNL extrusion set-up. The experimental results revealed that the orientation of PET nanofibers along the MD is a key factor that affects the tensile properties of PET nanofiber reinforced PP composites. The degree of nanofiber orientation along the MD can be controlled by the number of multipliers and mass flow rate used in the MNL extrusion process. As the multiphase polymeric melt flows through a multiplier, the dividing-multiplying process induces a stretching mechanism on the randomly dispersed high aspect ratio PET nanofibers enabling them to orient along the MD. A similar and more pronounced trend is observed when the same PP/PET blends are tested under increasing mass flow rates, where the PET nanofibers were shown to have a higher degree of orientation when the mass flow rate increased from 40 to 245 g/min. The synergistic effect of both parameters resulted in significant nanofiber alignment (i.e., lower average angles of orientation along the MD) by 83%, 68%, and 85% for 3, 7, and 15 wt.% PET cases, respectively. These effects were illustrated using the micrographs, showing the orientation development of the nanofibers from random to highly oriented under stronger processing conditions (i.e., higher number of multipliers and mass flow rates). The tensile properties, plotted with respect to processing conditions and angles of orientation, showed significant enhancements in elastic modulus and tensile strength, as a result of higher deformation ratio, shear stress, and thus higher degree of orientation of PET nanofibers along the MD, within each PET concentration cases. Overall, by integrating both nanofibrillation and MNL extrusion technologies, we can create a new class of multiphase polymer composites with tunable mechanical properties. Moreover, the MNL process has shown to exhibit a key advantage of efficient micro-/nanostructuring of multiphase composites to obtain enhanced properties.

## Figures and Tables

**Figure 1 polymers-12-02585-f001:**
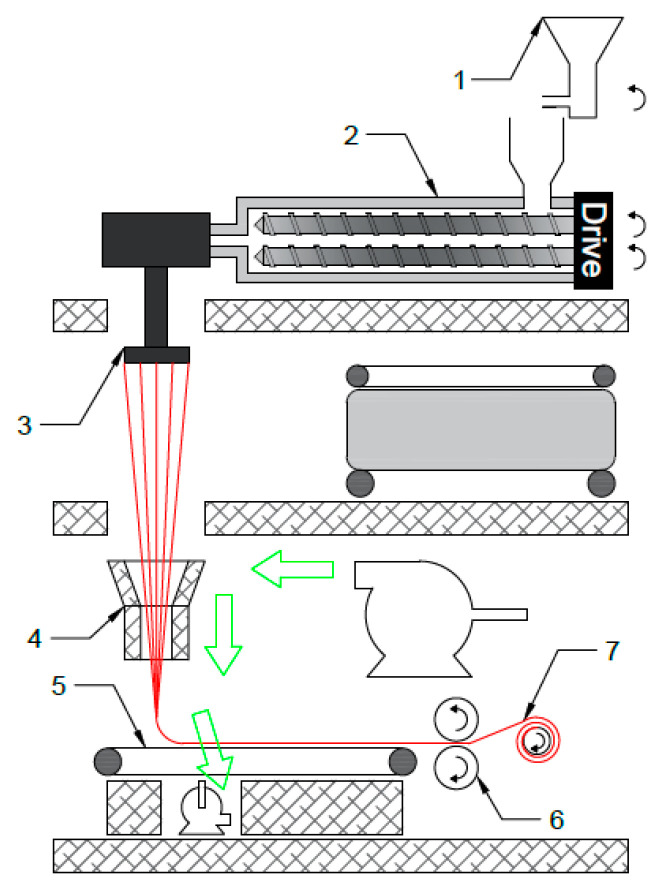
Schematic representation of a spunbond fibrillation system: (1) feeder, (2) twin screw extruder, (3) spinneret, (4) air drafter, (5) belt, (6) roller, (7) non-woven fiber mat.

**Figure 2 polymers-12-02585-f002:**
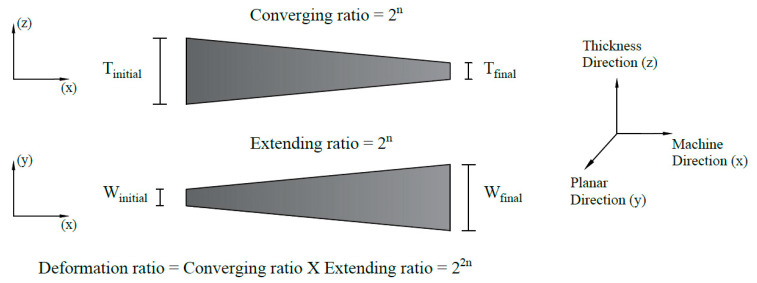
Converging and extending ratio of polyethylene terephthalate (PET) nanofibers as a result of multilayer flow.

**Figure 3 polymers-12-02585-f003:**
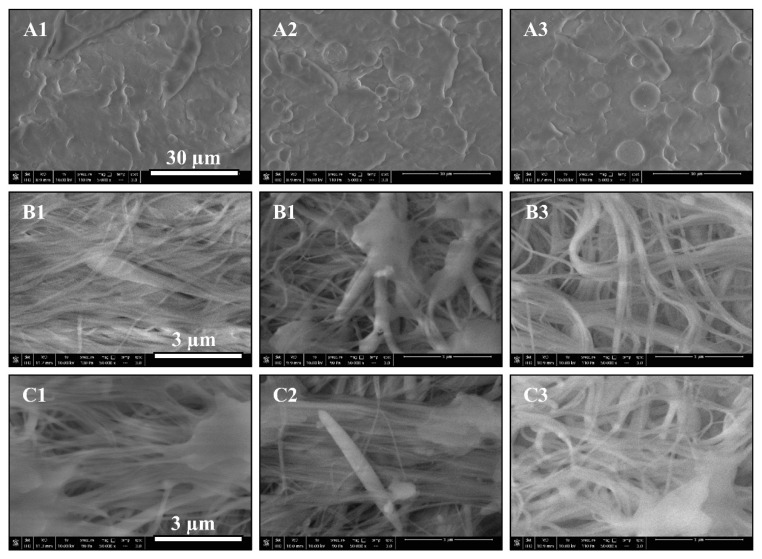
SEM images of the spherical and fibrillar PET dispersed in polypropylene (PP) matrix. 1, 2, and 3 denote PET concentrations of 3, 7, and 15 wt.%, respectively. (**A**–**C**) denote prespunbond, postspunbond, and post-MNL (micro-/nanolayer) processing stages, respectively.

**Figure 4 polymers-12-02585-f004:**
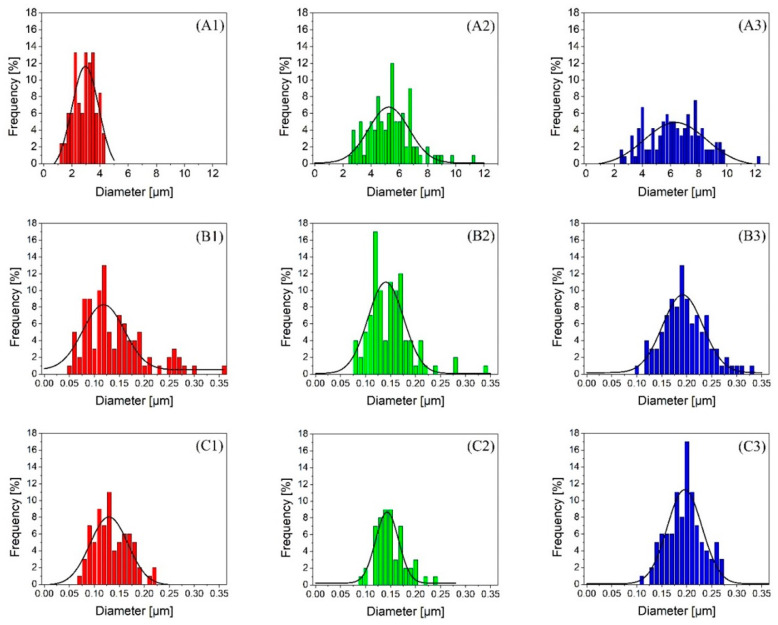
Size distributions of PET spherical and nanofiber diameters. 1, 2, and 3 denote PET concentrations of 3, 7, and 15 wt.%, respectively. (**A**–**C**) denote prespunbond, postspunbond, and post-MNL processing stages, respectively.

**Figure 5 polymers-12-02585-f005:**
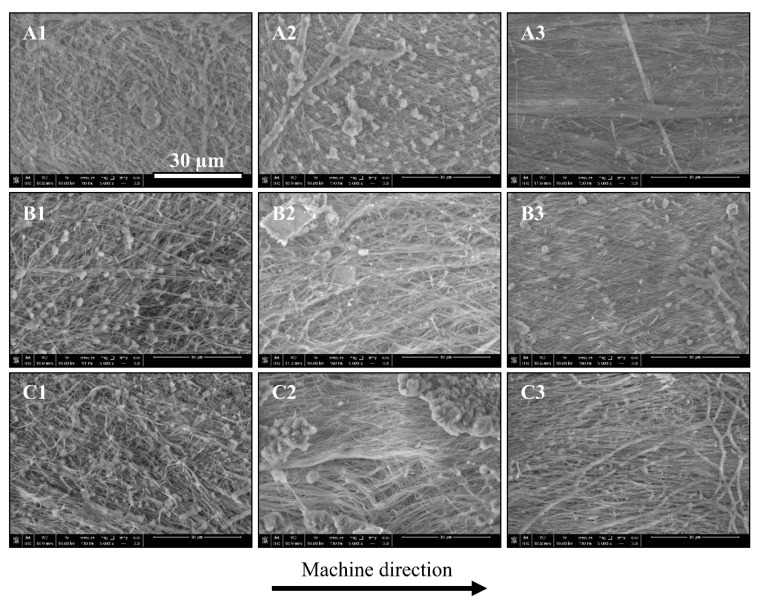
SEM images showing PET nanofiber orientation along the machine direction (MD). (**A**–**C**) denote PET concentrations of 3, 7, and 15 wt.% respectively. 1, 2, and 3 denote low extreme, intermediate, and high extreme processing conditions, respectively. All images were taken at 5000× magnification.

**Figure 6 polymers-12-02585-f006:**
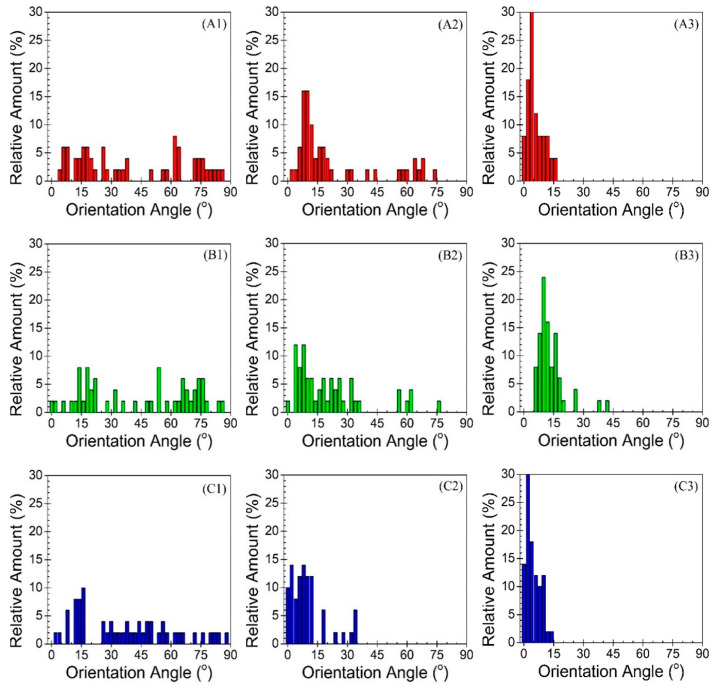
Distributions of PET nanofiber orientation angles with respect to the MD. 0° indicates perfect alignment along the MD, 90° indicates orientation of fibers transverse to the MD, and 45° implies random orientation. (**A**–**C**) denote PET concentrations of 3, 7, and 15 wt.%, respectively. 1, 2, and 3 denote low extreme, intermediate, and high extreme processing conditions, respectively.

**Figure 7 polymers-12-02585-f007:**
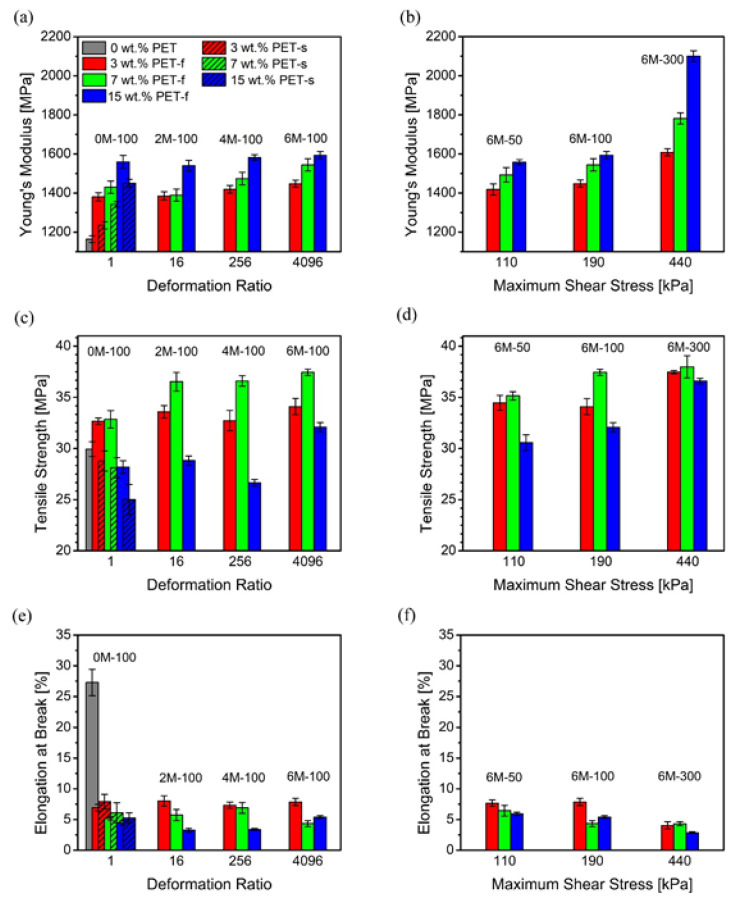
Tensile data of the MNL extruded PP/PET fibrillar blends. Charts (**a**,**c**,**e**) show the effect of number of multipliers (i.e., deformation ratio). Charts (**b**,**d**,**f**) show the effect of mass flow rate (i.e., shear stress). “f” and “s” denote fibrillar and spherical PET morphologies, respectively.

**Figure 8 polymers-12-02585-f008:**
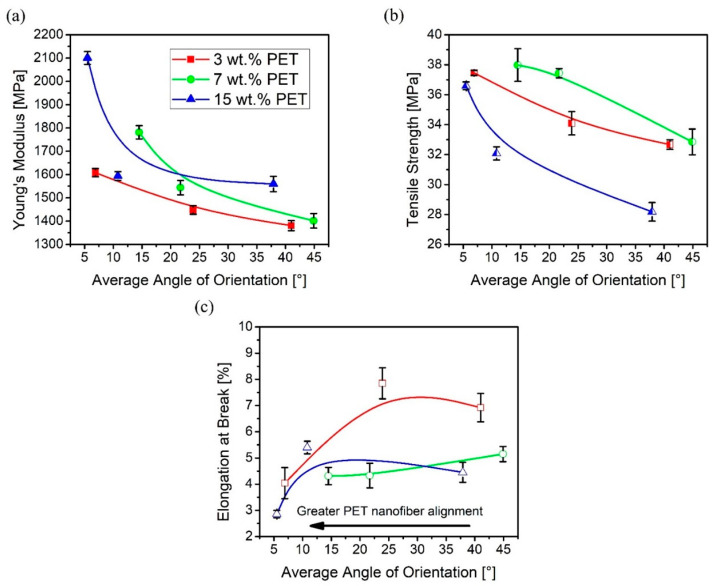
Variations of (**a**) elastic modulus, (**b**) tensile strength, and (**c**) elongation at break with respect to average angle of orientation along the MD. Solid lines are not trendlines but are presented to guide the eye.

**Table 1 polymers-12-02585-t001:** Sample codes and processing conditions.

	RPM	0 Multipliers	2 Multipliers	4 Multipliers	6 Multipliers
3 wt.% PET	50				3%-6M-50
	100	3%-0M-100	3%-2M-100	3%-4M-100	3%-6M-100
	300				3%-6M-300
7 wt.% PET	50				7%-6M-50
	100	7%-0M-100	7%-2M-100	7%-4M-100	7%-6M-100
	300				7%-6M-300
15 wt.% PET	50				15%-6M-50
	100	15%-0M-100	15%-2M-100	15%-4M-100	15%-6M-100
	300				15%-6M-300

**Table 2 polymers-12-02585-t002:** The average diameter and standard deviation of PET droplets and fibers under different PET concentrations and processing stages; “sph” denotes spherical PET, “fib1” denotes fibrillar PET in the postspunbond stage, and “fib2” denotes fibrillar PET in the post-MNL stage.

PET Concentration	Average Diameter (µm)	Standard Deviation (µm)
3 wt.%-sph	2.78	0.77
3 wt.%-fib1	0.14	0.06
3 wt.%-fib2	0.13	0.04
7 wt.%-sph	5.25	1.59
7 wt.%-fib1	0.14	0.03
7 wt.%-fib2	0.14	0.03
15 wt.%-sph	6.21	1.87
15 wt.%-fib1	0.19	0.05
15 wt.%-fib2	0.19	0.04

**Table 3 polymers-12-02585-t003:** Average angles of orientation and standard deviations of PET fibers along the MD.

PET Concentration	Processing Condition	Label	Average Orientation Angle (°)	Standard Deviation
3 wt.%	0M-100	A1	41.0	27.2
	6M-100	A2	23.9	21.4
	6M-300	A3	6.9	4.3
7 wt.%	0M-100	B1	44.9	26.4
	6M-100	B2	21.7	18.2
	6M-300	B3	14.5	7.0
15 wt.%	0M-100	C1	37.9	24.0
	6M-100	C2	10.8	9.1
	6M-300	C3	5.5	3.5

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
