# Peer review of "Tunable Tensile Properties of Polypropylene and Polyethylene Terephthalate Fibrillar Blends through Micro-/Nanolayered Extrusion Technology"

_polymers, 2020, doi:10.3390/polym12112585_

Round 1
Reviewer 1 Report
The manuscript describes the obtention of composites based on polymer blends whose mechanical properties can be finely tuned by the preparation procedure (the so called micro/nano-layered extrusion technology. The text is clear and concise and has the appropriate discussions for the obtained results. Hence, I support its publication in "Polymers" with minor changes, as stated below:
Is there any kind of information regarding the molecular weight and dispersity index (PDI) of the commercial-grade polymers used in this work? Could these parameters affect the obtained results?
Experimental conditions (scanning rate, pre-scan time and temperature, number of consecutive runs, and reproducibility) in which DSC data has been obtained should be specified. The same applies to rheology.
In section 2.5, I believe the figure that should be refereed is the number 5 and not the number 4.
Some of the appropriate labels for Figure 9 are missing.
Reviewer 2 Report
This manuscript is quite lengthy. Authors supplied many figures (12 figures) in which some of them are not necessary (For instance, Figure 3 and 6).
In addition, the organization of the entire manuscript is poor. It is completely wrong to present results (Figure 4 and 6) in the Methodology section.
Other comments are as follows.
Abstract – Problem statement is missing. Key findings are not highlighted at all!
Introduction – There is no even 1 reference for the first two paragraphs. The existing introduction is lengthy. It should be shortened by 25-30%. Besides, novelty and objective of this work should be clearly stated at the end of this section.
Section 2.2 – Authors should explain why such zone temperature range was chosen. Same goes to feeding frequency, screw rotating speed, and air drafter speed.
Figure 2 – I have no idea about what the image wants to deliver. It is lack of information!
Page 6 – Authors should not jump the figure number directly from Figure 5 to Figure 11!
Results and Discussion – Some parts of the discussion are too lengthy. Authors are strongly advised to provide brief but concise discussion.
Figure 9 – Images’ label is missing!!
Comparison with other findings reported in the literature should be carried out followed by discussion.
Round 2
Reviewer 2 Report
I'm now satisfied with the changes made by the authors.